# Artificial High Density Lipoprotein Nanoparticles in Cardiovascular Research

**DOI:** 10.3390/molecules24152829

**Published:** 2019-08-02

**Authors:** Karin Kornmueller, Ivan Vidakovic, Ruth Prassl

**Affiliations:** Gottfried Schatz Research Center for Cell Signaling, Metabolism and Aging, Biophysics, Medical University of Graz, Neue Stiftingtalstraße 6/IV, 8010 Graz, Austria

**Keywords:** lipoproteins, nanoparticle, reconstituted rHDL, apolipoprotein A1 peptide mimetics

## Abstract

Lipoproteins are endogenous nanoparticles which are the major transporter of fats and cholesterol in the human body. They play a key role in the regulatory mechanisms of cardiovascular events. Lipoproteins can be modified and manipulated to act as drug delivery systems or nanocarriers for contrast agents. In particular, high density lipoproteins (HDL), which are the smallest class of lipoproteins, can be synthetically engineered either as nascent HDL nanodiscs or spherical HDL nanoparticles. Reconstituted HDL (rHDL) particles are formed by self-assembly of various lipids and apolipoprotein AI (apo-AI). A variety of substances including drugs, nucleic acids, signal emitting molecules, or dyes can be loaded, making them efficient nanocarriers for therapeutic applications or medical diagnostics. This review provides an overview about synthesis techniques, physicochemical properties of rHDL nanoparticles, and structural determinants for rHDL function. We discuss recent developments utilizing either apo-AI or apo-AI mimetic peptides for the design of pharmaceutical rHDL formulations. Advantages, limitations, challenges, and prospects for clinical translation are evaluated with a special focus on promising strategies for the treatment and diagnosis of atherosclerosis and cardiovascular diseases.

## 1. Lipoprotein Metabolism and Coronary Heart Disease

Atherosclerosis is the major cause of morbidity and mortality in coronary heart diseases (CHD) and represents a substantial health and economic burden for the society. According to a report of the American Heart Association in 2019, cardiovascular diseases claim more lives each year than all forms of cancer and chronic lower respiratory diseases combined. In 2016 more than 17.6 million deaths were attributed to cardiovascular diseases (CVDs) globally [1]. These alarming numbers highlight the crucial need to gain a better understanding of cardiovascular events on a molecular level to enable the development of targeted therapeutic strategies. In particular, nanotechnology offers exciting new opportunities for the use of nanoparticles with specifically tailored features for targeted drug delivery and therapy. One promising option is to focus on biomimetic nanoparticles resembling naturally occurring nanoparticles, such as lipoproteins. Lipoproteins are the endogenous carriers of water insoluble compounds like fat or cholesterol in the aqueous environment of blood. Circulating lipoproteins are discrete nanoparticles that have well-defined structures and multiple biological functions playing a dominant role in lipid metabolism. Lipoproteins are key players in the progression—but also in the prevention and reversal—of atherosclerosis or cardiovascular events. High-density lipoprotein (HDL) particles are the smallest set of lipoprotein particles and possess mainly protective features against atherosclerosis. The underlying protective mechanism relies on the role of HDL in the removal of excess cholesterol from peripheral tissues, particularly from lipid-laden macrophages [2]. This process is known as reverse cholesterol transport (RCT) [3,4]. In brief, apolipoprotein AI (apo-AI), as the primary protein constituent of HDL, is synthesized predominantly in the liver and immediately associates with phospholipids and cholesterol by ATP binding cassette transporter A1 (ABCA1) dependent mechanisms. Nascent discoidal HDL particles of varying size and composition are formed by self-assembly. Discoidal HDL particles are lipid bilayer nanodiscs with diameters ranging from 7–13 nm and 4–5 nm in height, depending on the lipid composition [5]. For discoidal HDL it was suggested that two apo-AI molecules are wrapped around the lipid bilayer in a double belt-like helical manner, thus structurally stabilizing the lipid nanodisc [6]. The remodeling process of nascent HDL to spherical HDL proceeds by gathering free cholesterol, which is stored in the lipid bilayer of the nanodiscs. The free cholesterol is then converted by the enzyme lecithin cholesterol acyl transferase (LCAT) into cholesterol esters. Cholesterol esters are more hydrophobic than cholesterol and separate in an inner oily core. Consecutively the morphology of the discoidal HDL is altered to lens-like and finally to spherical particles, termed HDL3. Additional uptake of cholesterol and accumulation of enzymatically converted cholesterol esters transforms HDL3 to larger cholesterol-rich HDL2 particles. Such cholesterol-rich spherical HDL particles contain 2 to 4 apo-AI molecules in a similar helical but trefoil-like structural arrangement as observed in discoidal HDL [7]. With the help of cholesterol ester transfer proteins (CETP), HDL can exchange triglycerides and cholesterol esters with other lipoproteins such as very low density lipoprotein (VLDL) or low density lipoprotein (LDL). Subsequently, HDL delivers cholesterol to the liver for selective cholesterol uptake through scavenger receptor class B type I (SR-B1) mediated transfer in hepatocytes. Figure 1 shows a schematic illustration of HDL metabolism with focus on the RCT pathway depicting the metabolic conversion of HDL particles. For more detailed reviews on HDL metabolism, see references [3,8,9]. Apart from cholesterol transport, HDL particles act as carriers for vitamins, proteins or nucleic acids, thus playing a vital role in intercellular communication [10,11,12] Moreover, HDL shows anti-inflammatory and antioxidant properties by inhibiting lipoprotein oxidation by metabolizing lipid hydroperoxides [13,14]. These inherent features of HDL are extremely important considering artificial HDL particles as pharmaceutical nanocarriers. Naturally occurring HDL particles display an immense diversity and molecular complexity per se. Recent lipidomic and proteomic studies have identified up to 85 different proteins to be associated with HDL subspecies [15,16], while more than 200 individual lipid species are existing in native HDL particles [17]. In numerous studies an inverse correlation between blood HDL levels and the risk to develop CVD was found. This led to widespread acceptance that elevated serum concentrations of HDL are beneficial in preventing CVDs [18,19]. In keeping with these findings, many therapeutic approaches to increase the level of circulating HDL are constantly in the focus of pharmaceutical research [20,21]. However, it soon became clear that solely enhancing HDL cholesterol levels is far not enough for successful treatment of CVDs. Hence, there is a paradigm shift and recent strategies focus not only on approaches to increase the number of HDL particles in circulation but also to improve their functionality. One approach to improve functionality is to systematically research and exploit distinct structural features like size and morphology variations of HDL particles. However, the uniqueness of single HDL particles lies in their specific lipid and protein composition [22,23]. Given that the compositional and physicochemical parameter of HDL can be easily modulated and manipulated chemically, new research endeavors concentrate on the development of chemically modified or synthetic HDL particles as biomimetic systems [24,25]. Novel technologies for HDL-based applications include the use of engineered reconstituted HDL (rHDL) nanoparticles assembled from natural components of HDL i.e., phospholipids, cholesterol and apo-AI. Apo-AI is the most abundant protein in HDL that accounts for roughly 70% of HDL protein mass and about 50% of HDL total mass. Apo-AI is water-soluble protein, exchangeable between lipoprotein particles in solution and structurally characterized by a series of amphipathic α-helical repeats in its lipid binding domain [26]. The association of these amphipathic α-helical domains with lipids seems to be sufficient to create the morphology of the nanoparticle that resembles natural occurring discoidal nascent HDL particles. Thus, HDL particles can be conveniently assembled from individual lipids and protein to build up artificial HDL nanoparticles. To date, rHDL particles are well studied and present a highly attractive and flexible nanoparticle platform for medical applications. The potential of synthetic rHDL particles as nanocarriers in therapy and diagnostics is extensively reviewed by several groups [27,28,29,30,31]. A big advantage is that the biological behavior of rHDL particles can be fine-tuned by the choice of the phospholipid components used for the assembling process. Another key modulator of the distinct functions of rHDL in vivo is the structure of the surface associated protein moiety. In this respect, different structures of apo-AI including full-length apo-AI, truncated fragments, genetic variants of apo-AI like Apo AI_Milano_ have been widely explored [32,33]. More recent approaches focus on short chain apo-AI mimetic peptides belonging to class A amphipathic helical peptides [34,35]. Some pharmaceutically relevant examples that exploit the rHDL nanoparticle platform will be discussed later on in this review.

## 2. Synthesis of rHDL Nanoparticles

Artificially synthesized HDL mimetics need to meet certain physicochemical and biological criteria to be used for drug delivery and therapy. The nanoparticles need to have dimensions, density, and surface characteristics similar to HDL to be stable in circulation for longer times, and to show non-toxic, non-immunogenic and biocompatible properties [24]. 

Concerning their architecture, HDL mimetics can be either synthesized as phospholipid bilayer nanodiscs or as spherical lipid monolayer nanoparticles, both stabilized by apo-AI molecules. Spherical HDL nanoparticles reveal a core-shell arrangement composed of a lipophilic oily core surrounded by an outer amphipathic phospholipid monolayer which is surface-coated with proteins. This molecular arrangement enables a high loading capacity for poor water-soluble drug molecules in the inner core. In HDL nanodiscs apolar drugs are incorporated within the lipid bilayer. Amphipathic or hydrophilic drug molecules can be partially embedded into the phospholipid layer or attached to the surface. For in vivo applications, the nanoparticles need to show adequate pharmacokinetics and biodistribution, specificity to target tissues and cellular receptors in order to allow proper particle recognition, drug release or lipid exchange. To face these challenges, the first step is to establish reliable methods to produce high quantities of rHDL particles of excellent quality and batch-to-batch reproducibility. Traditionally, HDL particles are assembled from full-length apo-AI, either isolated from human plasma or engineered synthetically, and phospholipids. Wild type apo-AI can be directly isolated from human serum derived HDL particles, while recombinant apo-AI is usually expressed in Chinese hamster ovary (CHO) cells or *E. coli* [37,38]. For the synthesis procedure commercially available bilayer-forming phospholipids, e.g., saturated lipids like dimyristoyl-phosphatidylcholine (DMPC) or unsaturated lipids like palmitoyl-oleoly-phosphatidylcholine (POPC), and cholesterol are most frequently used. Typically, the lipid mixtures are dissolved in organic solvents and dried under a stream of nitrogen to form a thin lipid film. The dry lipid film is rehydrated with an aqueous buffer containing bile acid derivatives like sodium cholate as detergent together with the protein component. rHDL particles are readily formed by detergent-removal via dialysis against detergent free buffer. Alternatively, rHDL particles can be synthesized by solubilization of the phospholipid membranes of preformed liposomes using apo-AI [39]. Likewise, spherical rHDL particles can be prepared by sonication from lipid suspensions containing phospholipids and cholesteryl esters. Upon addition of apo-AI the protein associates to the lipid suspension to form spherical HDL [40]. An alternative approach to convert the discoidal rHDL nanoparticles into spherical particles is by incubation with LDL and LCAT to induce lipid exchange. To obtain more homogeneous preparations the particles are purified and recovered by gel chromatography or density gradient ultracentrifugation [41]. By varying the preparation conditions e.g., temperature, phospholipid composition, apolipoprotein species, molar lipid to protein ratio, or percentage of cholesterol added to the formulation, rHDL particles with tailored sizes can be rationally designed [42,43]. The most preferred lipid to protein ratio is about 100 to 1 mol/mol, yielding HDL particles with dimensions of about 10 nm. Larger discoidal particles of sizes up to 40 nm can be achieved by increasing the amount of lipids relative to protein [39,44]. Increasing the content of apo-AI in rHDL enhances its antioxidative activity towards oxidized LDL. In contrast, oxidation of a specific methionine residue (Met112) in apo-AI decreases its antioxidant activity but increases cholesterol efflux efficiency [45]. Hence many parameters concerning protein and lipid composition have to be considered for the design of rHDL. Naturally, HDL particles are composed of a crude mixture of phospholipid species, with PC and sphingomyelin (SM) being the predominant lipid classes. However, HDL also contains significant amounts of plasmalogens, lysophosphatidylcholines, phosphatidylethanolamine, phosphatidyl-inositols, and minor amounts of other negatively charged phospholipids [46,47]. Such lipid components can also be included in the synthesis of rHDL nanodiscs. Notably, the addition of negatively charged lipids like phosphatidylserine, phosphatidylglycerol, or cardiolipin for the synthesis of rHDL nanodiscs impacts not only HDL structure but also its biological function and therapeutic potential [47]. It is speculated that negatively charged phospholipids increase HDL binding to cell surface molecules by targeting specific phospholipid receptors [47]. Moreover, apo-AI preferentially associates with acidic lipid membranes via electrostatic interactions between lysine residues and negatively charged lipid headgroups [48], while anionic phospholipids likewise increase the α-helical content of apo-AI, which is important for receptor recognition [49]. More recent manufacturing techniques allow a large-scale production of great amounts of rHDL at continuous flow conditions in a single step procedure using microfluidic devices. By manipulating the mixing speed and the lipid to protein ratio the physicochemical characteristics like size and morphology of the HDL nanoparticles can be fine-tuned. A general scheme for a microfluidic setup for the production of rHDL particles is shown in Figure 2. 

A more specific device constructed by a three-channel inlet and a wide channel downstream of the inlets enables strong mixing through the formation of two micro-vortices. By adjusting the mixing speed and the lipid to protein ratio the physicochemical characteristics of the rHDL can be controlled. In particular, small rHDL nanoparticles of uniform size (~8 to 9 nm) can be synthesized in a single-step procedure at a production rate of up to 400 mg of particles per hour [50]. In this way, multifunctional rHDL particles can be engineered conveniently. The flexible microfluidics platform allows simultaneous loading of rHDL with signal emitting dyes or hydrophobic drugs like simvastatin. Likewise, inorganic gold, ironoxide, or quantum dot nanocrystals co-dispersed with lipids in organic solution can be incorporated as nanocrystalline core forming spherical rHDL nanoparticles for medical imaging (i.e., computer tomography (CT), magnetic resonance imaging (MRI), or fluorescence microscopy) [50]. Recently, hybrid rHDL particles containing a polymer or triglyceride core were developed as controlled drug release systems with adjustable sizes from about 10 to 100 nm depending on the core composition [51,52]. Artificial enrichment of rHDL particles with a polymer or triglyceride payload causes the conversion from discoidal rHDL nanodiscs to larger spherical rHDL particles, which can be straightforwardly loaded with fluorescent dyes or imaging agents for in vitro or in vivo screening. Currently, an rHDL inspired nanoparticle platform is established, thoroughly characterized and tested for its biological activity with the goal to develop tailored therapies for atherosclerosis and other inflammatory diseases [52]. Figure 3 provides a schematic overview of specific structures presenting the rHDL nanoparticle platform. 

## 3. Structural Aspects of apo-AI Mimetic Peptides

An attractive alternative for the synthesis of rHDL particles regards the use apo-AI mimetic peptides instead of full-length apo-AI. Such peptides possess amphipathic helix motifs similar to those of apo-AI or apolipoprotein E (apo-E) [53], but do not necessarily show sequence homology to these apolipoproteins. An indispensable requirement is that these peptide sequences show a high binding affinity to lipids in order to maintain their helical structure [54,55]. Over the last years, several clinically relevant apoA-I mimetics have been identified [56]. Conclusively, it was found that peptides with increased hydrophobicity (e.g., by substitution of the aliphatic amino acid residues by phenylalanine (F)) showed improved lipid affinity, while acetylation and amidation of the N- and C-terminus, respectively, helped to stabilize the peptide’s amphipathic α-helical conformation [57]. The most extensively studied group of peptide mimetics are 18 amino acids (18A) long and differ with respect to their number of phenylalanines, varying from 2 to 7 F residues. The F residues are presented on the non-polar face of the amphipathic helices, which is characterized by a high level of hydrophobicity. Due to its higher solubility in water 4F (Ac-DWFKAFYDKVAEKFKEAF-NH2) containing four F residues is the most investigated peptide [34,58], while the tandem dimer of 4F linked by a proline residue was shown to be even more effective than the monomer [59,60]. To improve peptide stability and to reduce proteolytic damage in vivo, 4F is frequently synthesized from D-amino acids [61,62]. Another lead peptide compound is a leucine (L) rich amphipathic peptide termed 22A [63]. 22A shows similar properties as assigned to 4F including reduction of pro-inflammatory oxidized lipoproteins and promotion of ABCA1-dependent cholesterol efflux in reverse cholesterol transport. A third class of helical peptides with the generic name ELK is composed only of glutamic acid (E), leucine (L), lysine (K), and optionally alanine (A). ELK peptides can be synthesized with varying degrees of amphipathicity and net charge. Like 4F and 22A, ELK peptides show a high efficiency in the induction of ABCA1-dependent cholesterol efflux [64]. 

Common structural motifs of apo-AI mimetic peptides that belong to class A amphipathic helices are shown in Figure 4. The amphipathic helices are arranged in a way that the hydrophobic and hydrophilic residues are located on opposing faces with the negative charges being aligned preferentially on the polar side [65,66,67]. This helical motif is attributed to interact with lipid membranes, thus stabilizing the membrane. 

Up to now various lipid–peptide complexes were synthesized with different lipid compositions as well as lipid to peptide ratios to create either discoidal or spherical rHDL particles. apo-AI peptide mimetics must have a strong binding affinity for lipids and the capacity to modulate lipid membranes [66]. Some of them bind to oxidized phospholipids and oxidized fatty acids even with a higher affinity than apoA-I determining their distinct anti-oxidative behavior [69,70]. However, to exert efficient anti-inflammatory properties lipid binding is crucial but not sufficient. By NMR studies it was shown that the association of the secondary structural motifs with the lipids impact the biological functions of the peptide. It was suggested that the proper arrangement of the aromatic residues on the non-polar phase is more important for the anti-inflammatory action of the peptide than the overall hydrophobicity and that the peptides should be loosely associated with the lipid head groups and not tightly bound to the acyl chains [71,72]. Thus, subtle changes in the physicochemical characteristics and the structural arrangement of the peptides might influence their biological activity in vivo [73]. Recent advances in the development of apo-AI peptide mimetics and applications of the free peptides as therapeutics for the management of atherosclerosis and CVDs are reviewed in detail elsewhere [74,75,76]. In this review we focus on the use of apo-AI mimetics for the design of rHDL particles. 

## 4. Physicochemical Characterization of rHDL

A thorough chemical and biophysical description of rHDL particles is required as quality control measure. In particular, for modified or multifunctional particles the chemical composition has to be validated following purification steps. For the quantification of the phospholipid, protein, triglyceride, or cholesterol content various commercially-available assays are available, which are normally based on a colorimetric method. In case of labeled rHDL particles care has to be taken not to interfere with the fluorescent signal of the marker molecules. Labels consisting of inorganic molecules like ironoxide- or gold nanoparticles can be analyzed by inductively coupled plasma mass spectrometry (ICP-MS). To determine the payload of rHDL particles different analytical methods are available depending on the loaded drug molecules. Routinely, the drugs are extracted by organic solvents and analyzed with high performance liquid chromatography (HPLC).

For the biophysical characterization the structure, morphology, and molecular organization of rHDL particles can be measured by various complementary techniques including analytical ultracentifugation, size exclusion chromatography (SEC), native polyacrylamide gel electrophoresis (native PAGE), agarose gel electrophoresis, dynamic light scattering (DLS), nanoparticle tracking analysis (NTA), transmission electron microscopy (TEM), cryo-TEM, atomic force microscopy (AFM), and X-ray or neutron scattering techniques. Each of these techniques has its advantages and disadvantages and only a combination of several techniques permits a reliable classification of the rHDL based nanoparticles. Human plasma HDL particles are characterized by their inherent heterogeneity in lipid composition as well as in their chemical, structural, and biological characteristics [77]. Upon analytical ultracentrifugation HDL particles can be separated in distinct subclasses distinguished by their density and lipid composition. Analytical ultracentrifugation is highly accurate but needs expensive installations, is time consuming and needs expert skills for interpretation [78]. In case of rHDL, the recombinant preparations should be homogeneous in terms of lipid composition and lipid distribution, while the polydispersity is given as size heterogeneity related to the self-assembling process. 

A classical method for particle size determination is DLS. The technique is based on Brownian motion of single particles in solution and allows the determination of the hydrodynamic radius of the particles. The polydispersity index is provided as an estimate for the heterogeneity of the samples. As an average particle size is determined from the intensity of the scattered light of all particles measured for a certain time, larger particles have a higher contribution to the signal than small particles. Hence, a very low amount of large particles or aggregates may falsify the results. Moreover, the lower limit for particle size determination by DLS is about 10–20 nm making it difficult to determine the size of smaller rHDL particles. NTA operates similar to DLS measuring the Brownian motion of single particles in a small observation window by laser scattering. The advantage of NTA is that the scattered light of the nanoparticles can be seen directly with a microscope attached to the instrument. From the defined volume the number of particles in solution can be estimated and the polydispersity of the preparation is given as the percentage of particles (i.e., D10, D50, or D90) which are below a certain size. The limitations concerning the size determination of smaller particles are similar to DLS. An interesting approach for single particle tracking of lipoproteins was described recently using fluorescent labeled lipoproteins to be tracked by wide-field fluorescence microscopy [79]. With this technique the motion of very small HDL particles of about 5 nm could be tracked. For NTA measurements the rHDL samples have to be diluted to a proper concentration corresponding of about 107 to 109 particles/mL to enable single particle tracking. Hence, only about 100 individual particles can be tracked per single measurement, which makes multiple repetitions indispensable. Compared to DLS the error bars of the size distribution obtained by NTA are larger as a consequence of the variation in particle counts between the measurements and the fact that a lower amount of data are available for statistical evaluation. Moreover, NTA requires several optimization steps including dilution series and proper settings for video capture and analysis [80]. For quality control measurements NTA is more time consuming than DLS, but enables the visualization of the individual particles and the presence of few large particles does not affect the sizing accuracy [80]. Despite subtle differences in the results obtained by measuring either bulk solutions or single particles with DLS and NTA, respectively, both techniques are comparable and suitable for size determination. Preferentially, they should be combined following standardized protocols for quality assurance. With SEC different size populations can be separated and detected independently. SEC also allows to remove impurities and to monitor the labeling or loading efficiency. For instance, by co-elution of radioactivity with the nanoparticles proper labeling quality is assured [81]. Once calibrated with commercially available standards the Stokes radius of the rHDL particles can be estimated. Native PAGE is very well suited to determine the molecular size and homogeneity of the particles using a high molecular weight protein standard as reference. The gels can be stained with Coomassie blue or Sypro Ruby for protein and with Sudan black for the phospholipids. Native PAGE is a very sensitive technique to evaluate particle size distributions, however, this method is labor intensive and difficult to be standardized between laboratories. The techniques discussed so far are well suited for the characterization of physicochemical properties of rHDL particles, but with none of them it is possible to distinguish between spherical or discoidal particles. Thus, the most prominent technique to determine both particle size and morphology is negative staining TEM. For TEM measurements a small drop of the rHDL particles is placed on a copper grid, blotted and stained with either uranyl acetate or phosphotungstic acid. In TEM discoidal HDL particles are typically seen as stacked nanoparticles resembling a “rouleaux” formation, however, it is speculated that the drying process causes some aggregation artefacts seen as nanodisc stacking in the TEM images [43,82]. To overcome this problem Zhang et al. provided an optimized protocol for negative staining TEM of HDL particles [82]. Exemplary TEM images of plasma HDL and discoidal rHDL particles reconstituted from DMPC/cholesterol and either full length apo-AI or apo-AI peptidomimetics (4F) at molar protein/peptide to lipid ratios of 1/2 to 80 are presented in Figure 5. 

Drying processes could also cause some shrinking of the particles. This shortcoming can be overcome by using cryo-EM. In cryo-EM the rHDL particles are shock-frozen from solution in a fully hydrated state without any chemical fixation or staining [83]. The particles can be imaged directly in vitreous ice. As the phospholipid headgroups have a low contrast comparable to the solvent the protein component becomes clearly visible at the surface of the particle. A more recent cryo-EM study using a homogeneous fraction of large rHDL particles, which are composed of full length apo-AI, cholesterol and excess of DMPC (1:74:423 mol/mol), clearly shows discoidal single particles of rHDL. Selected two-dimension class averaged images were used for 3D reconstruction showing dimensions of rHDL particles of 36 nm diameter and 4.5 nm height. The height corresponds to the expected thickness of a single phospholipid bilayer [44]. Considering recent technological advances in single particle cryo-EM including imaging and software developments [84], and the constantly increasing availability of the technique in many laboratories, cryo-EM will most probably become the gold standard for structural investigations of reconstituted lipoproteins. Small angle X-ray (SAXS) and neutron scattering (SANS) techniques are alternative tools to obtain valuable structural information on rHDL particles. SAXS is an excellent technique for the delineation of overall size and shape of rHDL particles, while SANS with contrast variation provides detailed information on the structural arrangement of protein and lipids within the particle [85,86]. However, both techniques are not easily accessible and certainly need experts for data evaluation. Figure 6 provides an overview of the most prominently used techniques for the physicochemical characterization of rHDL particles and the information obtained from it. 

## 5. rHDL-Based Therapy in CVDs

A vast number of studies describe the direct use of rHDL as therapeutic agent addressing the central anti-atherogenic and cardioprotective properties of HDL [87]. Repeated infusion treatment with rHDL was shown to reduce inflammatory and atherogenic processes in different animal studies [88]. As an example, in a rabbit model of vascular inflammation a daily infusion of rHDL composed of dipalmitoyl-phosphatidylcholine (DPPC) and apo-AI successfully inhibited neutrophil infiltration and the expression of adhesion molecules [89]. Supplementation or rHDL with a negatively charged phospholipid, i.e., phosphatidylserine (PS), reduced the plasma levels of inflammatory biomarkers including IL-6 or TNFα, and intercellular adhesion molecules in a mouse model of atherosclerosis. Interaction with PS specific receptors could contribute to the anti-inflammatory action of PS containing rHDL particles [90]. Applied in vivo rHDL particles are subject to intravascular remodeling upon interaction with apolipoproteins and lipid transfer proteins (recently reviewed by Darabi and Kontush [91]). From in vitro and pre-clinical studies using various different lipid species for the synthesis of rHDL nanodiscs it became obvious that the lipid composition plays an intriguing role in HDL triggered signal transduction. Apart from this, the lipid composition is a determinant for rHDL size, charge, structure, and stability as outlined before. rHDL composition and structure further determines cholesterol efflux capacity and interaction with hydroperoxides [45]. 

Encouraged by promising preclinical studies, human clinical trials were designed revealing that rHDL is well tolerated by patients. It was found that intravenous injections of rHDL significantly raise the plasma levels of HDL in plasma of humans [31,47]. HDL raising therapy is expected to promote RCT resulting in an inhibition of fatty streak formation and lipid deposition in atherosclerotic walls as well as a reduction of atheroma volume and plaque remodeling [92]. Currently, several rHDL therapeutics are in clinical evaluation. As an example, CSL-112 or CSL-111, the predecessor compound, are nanoassemblies of soybean phosphatidylcholine with wild type apo-AI at different lipid to protein stoichiometries, 75:1:1 and 150:1 for CSL-112 and CSL111, respectively, with a heterogeneous size distribution ranging from 7–30 nm in diameter [93,94]. CSL-112 given as single infusion to patients with stable atherosclerotic disease immediately raised apo-AI levels and dramatically increased ABCA1 dependent cholesterol efflux [94,95,96]. Once in circulation the rHDL particles are rapidly remodeled into mature spherical HDLs [97]. A potential recycling of apo-AI from rHDL into the endogenous pool of HDL particles contributed to longer circulation times than achieved by infusion therapy using native HDL [97]. CER-001 from Cerenis Therapeutics is another example of an HDL raising agent that reached clinical phase II. This formulation is assembled from recombinant apo-AI, SM and dipalmitoyl-sn-glycero-3-phosphorylglycerol (DPPG) at a ratio of 1:2.7:0.1 mol/mol. The rHDL nanoparticles are in a size range between 7–13 nm and negatively charged. The negative surface charge reduces the fusion tendency of these particles with endogenous lipoproteins and significantly contributes to the stability of the particles in circulation [98]. The formulation shows favorable effects on cholesterol efflux and vascular inflammation [99], however, CER-001 infusions did not lead to the regression of coronary atherosclerosis i.e., measures of coronary plaque volume as primary endpoint in patients with post-acute coronary syndrome and high plaque burden [100]. These rather disappointing results on CER-001 point to the fact that apart from increasing the number of HDL particles and HDL cholesterol through HDL infusion therapy, HDL structure and dynamics have to be taken into account and should be considered more seriously [31]. Details on the clinical performance of these agents are reviewed elsewhere [28,29,101]. For the optimization of rHDL based formulations the inclusion of other lipid species with enhanced biological activity and improved cellular specificity have been discussed [102]. Moreover, for clinical translation of rHDL formulations particular attention has to be paid not only on structure–function relationships but also on fast and reliably manufacturing procedures and production costs [47]. 

## 6. rHDL as Delivery Systems in Therapy and Diagnosis of CVDs

rHDL are highly attractive entities to be exploited as delivery systems for drugs with low water solubility. The drugs can be embedded either in the amphiphilic lipid layer of nanodiscs or loaded in the oily inner core of spherical particles. Within the lipid microenvironment of rHDL particles huge amounts of drugs can be transported in plasma being shielded from rapid degradation and immune attack. To achieve efficient delivery of the drugs to the target site rHDL need to have long circulation times and enhanced plasma stability. While it has been shown that apolipoproteins circulate for 2–3 days, the lipids and fats have a much shorter half-life. With the aim to improve circulation half-life a recent article by Pownall et al. highlights some interesting points concerning the lipid composition [102]. First, it is important that the lipids are resistant to phospholipases. This can be achieved by replacing ester bonds by non-hydrolyzable ether bonds. Second, saturated long acyl chain lipids should be preferred for the synthesis to make rHDL particles more lipophilic and less prone to oxidation. In fact, rHDL have some benefits over other delivery systems like liposomes or polymeric nanoparticles. They are extremely small and have the possibility to penetrate into tissues to some extent [103]. In particular, the leaky microvasculature of atherosclerotic lesions enables the infiltration of HDL particles into the intima [104,105,106]. Through this mechanism, rHDL particles can be retained in the plaque microenvironment for longer time periods. Other benefits of rHDL particles are their intrinsic targeting properties, which are provided via recognition by specific cell surface receptors expressed on target tissues [107,108]. Site specific recognition of rHDL by SRB1 or ABCA1/ABCG1 allows the direct delivery of lipophilic therapeutics into the cytosol [109,110,111]. Thus, the drugs can be transported across the plasma membrane to target intracellular sites. The inherent targeting capacity of rHDL implies that there is no need to conjugate an additional targeting ligand to the particle surface. One exception is when rHDL particles should be redirected towards other biomarkers than their natural receptors. For example, rerouting can be achieved by covalent coupling of folate to rHDL with the intention to address folic acid receptors, which can be overexpressed in activated macrophages in atherosclerotic lesions or in some tumor microenvironments [112,113,114]. Lastly, rHDL resemble natural endogenous HDL particles they have the chance to evade elimination by the mononuclear phagocytic system (MPS). 

Apart from drug delivery rHDL particles can be engineered as carriers of contrast agents for diagnostic imaging. In CVDs there is urgent need to recognize atherosclerotic lesions at early time points and to differentiate between stable and vulnerable plaques. The latter are prone to rupture causing stroke or myocardial infarction. Vulnerable plaques are generally identified by a thin fibrous cap, strong neovascularization and a higher amount of monocyte derived macrophages, especially in the area of the plaque shoulder. Along this line, rHDLs have been demonstrated to target lipid-laden macrophages in atherosclerosis, and due to their small size they can penetrate the plaque, whereas macrophage infiltration is positively correlated with lesion progression, plaque size, and intimal thickness. Much effort has been made in controlling the size of rHDL particles, a feature critically important for delivery of payload to the cytoplasm and avoidance of sequestration by the endosomal system [115]. Shaish et al. [116] were the first to show an accumulation of radiolabeled rHDL in atherosclerotic lesions of the aortic arch in apo E-/- mice. Based on these very promising results, Cormode and colleagues have continued to adopt HDL for contrast enhanced MRI [117,118,119]. For MRI, gadolinium (Gd-) based contrast agents are incorporated into rHDL particles which are reconstituted from single lipid components enriched with a chelate carrying lipid [118,120,121,122]. The most frequently used chelator molecule is 1,2 di-myristoyl-sn-glycero-3-phosphatidylethanolamine–diethylenetriamine pentaacetic acid (DMPE-DTPA) that provides the possibility to complex free gadolinium ions, which are otherwise toxic. Amphiphilic fluorescent dyes like Rhodamine B or Cy5.5 labeled PE-lipids can be added in the phospholipid layer. The labeling of rHDL with dual probes for MRI and optical imaging allows multimodal imaging in vivo as well as ex vivo. e.g., at 24 h post-injection rHDL nanoparticles are accumulated in the macrophage rich domains of aortic plaques in apo E-/- mice showing substantially enhanced MRI signaling [121]. Similar results were achieved when using discoidal rHDL particles instead of spherical ones [123]. Later on, the DMPE-DTPA lipid was substituted by an AAZTA-(1,4-bis(hydroxycarbonylmethyl)-6-[bis(hydroxycarbonylmethyl)] amino-6-methylperhydro-1,4-diazepine) -lipid. These particles revealed a higher relaxivity than the Gd-DTPA containing rHDL, making them even more efficacious contrast agents [124,125]. Over the last decade, the research groups around Cormode, Mulder, Fayad and colleagues have further refined the imaging platform by producing fully synthetic rHDL particles containing spherical nanocrystalline cores (reviewed by [27,36,126,127]). Generally, the inner lipophilic core is surrounded by a monolayer of phospholipids with apo-AI molecules associated to the surface. The nanocrystalline core is composed of either Au-nanoparticles (Au-NPs) for CT imaging, quantum dots (QD) for optical imaging [128], or ironoxide-nanoparticles for MRI [128,129]. Additionally, fluorescent dyes or phospholipid-anchored Gd-chelates can be incorporated into the outer phospholipid monolayer of the fully synthetic rHDLs to construct a multimodal signal-emitting particle for molecular imaging. The rHDL particles containing in inorganic core are spherical having sizes in the range between 7–12 nm as determined by TEM. They are highly homogeneous and have a lipid to protein ratio similar to natural HDL [128]. In vivo experiments in the apo E-/- mouse model for atherosclerosis revealed an accumulation of nanocrystalline rHDL in the vessel walls and in macrophages of atherosclerotic plaques [128]. Only recently, the biomimetic synthetic HDLs containing a gold nanocrystalline core have proven successful in the treatment of lymphoma by inhibiting B-cell lymphoma growth through binding to SR-B1, cholesterol starvation and selective induction of apoptosis [130]. rHDL mimics bearing a gold core surrounded by a phospholipid bilayer and apo-AI show efficient cellular cholesterol efflux from model cell lines and primary macrophages. Such combination systems provide the opportunity for infusion therapy enhancing macrophage cholesterol efflux via known pathways in addition to imaging [131]. In an alternative approach, McMahon et al. [132,133] have explored synthetic HDL with a gold nanocrystalline core for gene delivery. Here, Au-NPs were mixed with apo-AI in aqueous solution prior to assembly with lipids. Finally, cholesterol modified DNA was absorbed to the surface to form hybrid chol-DNA-HDL AuNPs. This conjugate was internalized by human cells and efficient cellular nuclear acid delivery was achieved. Overall, this study combines for the first-time lipid-based nucleic acid transfection strategies with HDL biomimicry for cell-specific targeting. In the following, AuNP core functionalized rHDL were also successful in the delivery of therapeutic short interference RNA (siRNA) [134,135]. Similarly, rHDL particles showed high potential for the delivery of microRNA (miRNA) for gene silencing and for the modification of gene expression profiles [10,11,12]. Native HDL have been identified to function as endogenous carriers for miRNAs in circulation and as transporter for miRNAs to recipient cells in the setting of intercellular communication [11,136]. These observations made rHDL delivery systems for miRNA/siRNA therapeutics even more interesting for gene therapy [10]. Detailed background information on the current challenges of nucleic acid therapy and future strategies to optimize HDL as transporter for nucleic acids is given in recent reviews [28,134]. A considerable number of studies have been conducted on rHDL as drug delivery system for small molecules, most of them dealing with cancer chemotherapy (see some recent topical reviews from the Lacko group [107,110,136]). In CVDs, a promising strategy is to target inflammation, which is a main feature in atherosclerosis. In this respect, Duivenvoorden et al. have recently developed statin loaded rHDL particles to deliver simvastatins to atherosclerotic plaques to inhibit inflammation [137]. The anti-inflammatory effect of the statin-rHDL particles was observed in vitro by a decreased survival of macrophages, and in vivo in an Apo E-/- mouse model of atherosclerosis. After one-week high dose treatment a significant reduction in inflammation was observed with MRI using Gd-labeled particles, while a 3-months low dose treatment inhibited the progression of plaque inflammation by direct targeting of plaque macrophages [137,138]. These highly promising results lead the authors to conclude that the anti-proliferative treatment strategy using rHDL in combination with lipid-lowering drugs has opened a new therapeutic window to address inflammation in atherosclerosis. In another example, Zhang et al. have loaded tanshinone IIA, a vasoactive cardioprotective drug, in discoidal and spherical rHDL particles. In vivo, the discoidal rHDL was enzymatically remodeled to spherical rHDL and tanshinone IIA was delivered to foam cells in atherosclerotic lesions irrespectively of which rHDL particle was used. Both rHDL particles showed strong anti-atherogenic efficacies in New Zealand white rabbits used as animal model [139,140]. 

Given the recent success, the next frontier in rHDL based strategies will be the optimization of multifunctional rHDL particles that integrate both, imaging and therapeutic agents within one single particle. In cardiovascular research, these particles might act as “nanotheranostics” for detection and imaging of early atherosclerotic plaques simultaneously acting as therapeutics to inhibit plaque growth [101,141]. 

## 7. Apo-AI Mimetics in the Therapy of CVDs

Among the various approaches of modulating the structure of HDL, the most promising alternative strategy is to use peptidomimetics instead of full length apo-AI. The peptide mimetics are biologically active through inhibiting inflammation and atherosclerosis by promoting reverse cholesterol transfer [76], and by sequestration and removal of oxidized lipids [34]. Due to their anti-inflammatory and anti-oxidant characteristics apoA-I mimetics are considered as therapeutics for the treatment of a range of diseases [35]. Hence, numerous studies exploring various peptide sequences report on the beneficial properties of peptidomimetics in vitro and in preclinical animal studies [142]. Although the peptides revealed excellent performance in animal models of atherosclerosis, they did not show efficient activity in early human trials [69,143]. Now, the peptides have been improved and recent observations have pointed to promising therapies for patients with CVD. The peptides appear to be well tolerated and effective in promoting the anti-inflammatory properties of HDL [144]. In early human studies it was found that 4F has additional benefits in patients taking lipid-lowering drugs like statins [32]. Other studies report on apoA-I mimetics as potential therapeutics for the treatment of obesity and the reduction of adipose tissues [145,146,147]. Further beneficial effects attributed to apoA-I mimetics include an amelioration of insulin resistance, increase in adiponectin levels and reduction of oxidative stress in chronic kidney disease [148]. The anti-atherogenic properties of the apo-AI mimetics are mainly attributed to their high binding affinity to ABCA1 transporter promoting reverse cholesterol transfer by mediating the activation of JAK-2 in macrophages [59], while their anti-inflammatory action is most likely mediated through binding to oxidized lipids attenuating the production of pro-inflammatory cytokines [61,69,149,150]. Apart from infusion therapy, apo-AI mimetics were optimized for oral applications. Some examples of promising pre-clinical studies on apoA-I mimetics applied orally include the peptide 6F that revealed potent anti-inflammatory, anti-oxidant, and anti-atherogenic effects in LDL receptor null mice [151]. It was shown that the peptides act on the small intestine to prevent systemic inflammation and dyslipidemia caused by a Western diet [152,153]. After oral administration of D-4F a synergistic anti-inflammatory effect with statins was observed in mice and monkeys [154]. In a first study performed in patients with high cardiovascular risk a single dose treatment with the peptide D-4F applied orally was safe and rendered HDL less inflammatory, however, the bioavailability of the free peptide was low [155]. In a follow-up randomized controlled trial the effects of multiple doses of oral D-4F were tested. The daily oral peptide dosing was well tolerated, and the HDL inflammatory index was effectively reduced compared to placebo treatment [156]. As mechanism of action it was suggested that apo-AI mimetics require sufficient enteric delivery to prevent villus inflammation and perhaps drive transintestinal cholesterol efflux [156]. Taken together, these results show the prospect of apo-AI mimetics for the treatment of CVDs with the aim to improve the function of HDL. 

## 8. rHDL Apo-AI Mimetics in the Therapy of CVDs

As mentioned above, recent efforts in developing artificial rHDLs have been expanded to include peptide mimetics instead of full length apo-AI to mimic the natural behavior HDL. To this end, various lipid–peptide complexes were synthesized with different lipid compositions as well as lipid to peptide ratios [71]. Typically, apo-AI mimetic peptides bind to lipids to form nanodisc HDL particles that were able to reduce atherosclerosis by removing cellular cholesterol and phospholipids through ABCA1 transporter dependent mechanisms similar to native HDL [64,157,158]. A wide range of peptides with different net charges and hydrophobicities were tested to show that a subtle balance between structural stabilization of rHDL nanodiscs, lipid binding capacity, and cytotoxicity exists. If the peptides were too hydrophobic, they extract lipids in an uncontrolled ABCA1 independent way to become toxic to cells [65]. Equally important is the phospholipid composition of rHDL nanodiscs. Schwendeman et al. formulated 5A either with lipid POPC or SM [159]. After injection of the rHDLs in rats it was found that 5A-SM was much more effective in mobilization of plasma cholesterol. While both rHDLs induced HDL remodeling after incubation with plasma and exhibited anti-inflammatory properties, 5A-SM showed a higher inhibition of cytokine release than did 5A-POPC. Zhao et al. used a multimeric branched peptide for the manufacturing of rHDL [160]. The multivalent peptides very highly stable towards proteolytic digestion and revealed long half-life in LDL receptor-null mice. The efficiencies of the monomer and the timer both formulated with DMPC were compared to each other. Surprisingly, the rHDL formulations administered ip were similar effective in lowering plasma cholesterol and reducing the development of atherosclerosis. When applied orally, both formulations were active, despite having undetectable plasma concentrations. The authors conclude that the mechanisms of the anti-atherogenic activity of the rHDL peptides are distinct different, one based on the plasma compartment and the other based on the gastrointestinal tract [161]. This finding is in agreement with other studies that suggest that the intestine is the primary site of action for apo-AI peptides [150,161]. The first apo-AI mimetic peptide formulation that was translated into clinical evaluation was the synthetic lipid formulation named ETC-642 composed of the peptide 22A and a mixture of SM and DPPC at 1/1/1 weight/weight ratio. [162,163]. ETC-642 was able to reduce the endothelial expression of intercellular cell adhesion (ICAM-1) and vascular cell adhesion molecule (VCAM-1) in the vascular tissues. A single low dose intravenous infusion of ETC-642 was very well tolerated by patients with stable coronary artery disease [164]. The results of a second multiple dose clinical study have not been published and the development of ETC-642 was terminated since Pfizer decided to stop the development of drugs for heart treatment based on the failure of torcetrapib in phase III clinical trial [165].

Apo-AI mimetic peptide based rHDL particles have also been widely explored for imaging of atherosclerotic plaques. As an example, a synthetic HDL nanodisc carrying the apo-AI mimetic peptide 37pA was functionalized with Gd-chelate complexes and rhodamine for combined MRI and fluorescence imaging, respectively. A comparison between apo E-/- mice and wild type mice after in vivo administration of the functionalized HDL nanodiscs revealed a significant enhancement of the MRI signal in the aortic plaques of the apo E-/- mouse but no significant signal in aortas of wild type mice [117]. By confocal fluorescence microscopy it was confirmed that the HDL nanodiscs preferentially accumulate in macrophage-rich areas of the atherosclerotic lesions. The same group showed that the targeting potential could be further enhanced using an apolipoprotein E (apo-E) derived lipopeptide, termed P2fA2 [166]. P2fA2 is a cationic arginine rich dimeric peptide that shows remarkable membrane penetrating properties. To increase the lipophilicity of the peptide it was conjugated with two fatty acid chains to create a cationic lipopeptide that could be easily incorporated into the phospholipid layer of rHDL. The P2fA2 enriched Gd-bearing rHDL nanoparticles showed significant signal enhancement of the atherosclerotic wall in MRI. The particles primarily co-localize with intraplaque macrophages as seen with CLSM, confirming that the peptides increase the uptake of modified HDL particles by macrophages in plaque regions [166]. However, these are only a few examples that point to the applicability of rHDL peptide mimetics for molecular imaging of atherosclerosis. For further details the reader is referred to several excellent reviews by the group around Mulder et al. [81,118,119,167]. Besides imaging and therapy of CVDs, the peptide based rHDL particle platform shows great promise for the treatment of a variety of infectious or inflammatory diseases, most relevant for cancer therapy [110]. Table 1 shows some representative examples of therapeutic rHDL nanoparticles tested in either preclinical or clinical settings.

Despite constant improvements, much more work needs to be done to achieve a better understanding of structure-function relationships of the differently assembled artificial rHDL particles to enable a more rational design of highly efficient apo-AI peptide based rHDL therapeutics.

## 9. Challenges, Advantages, and Limitations

rHDL particles have shown great promise as a diagnostic and therapeutic entity in cultured cell and animal models. However, given the broad range of research studies on numerous rHDL formulations, only a few of them have reached early clinical phase trials [168,169]. The translation of novel formulations from the laboratory to the clinic, especially dealing with nanosized particles, faces several challenges before the market is approached in a realistic timeframe of 10–12 years (Figure 7).

One challenge in the translation from the laboratory to the clinic among others is certainly the need for large scale production at good manufacturing practice (cGMP) conditions. When apo-AI is purified from human plasma to create HDL mimetics, as it is currently the case for CSL111 and CSL112, high quality criteria are required to ensure that the plasma product is free from infectious or immunogenic impurities. If recombinant proteins are used multi-step processes in expression and purification are required, which are rather time consuming and costly. Given these shortcomings, the advantages of fully synthetic products over naturally derived components are lower costs for production and higher flexibility in modifying the structure of rHDL. However, to enable translation scalability of manufacturing measures like yields and sample purity have also to be considered [101,170]. In this respect, the implementation of apo-AI peptide mimetics holds significant promise. The synthesis of short peptide sequences is less expensive than the production of apo-AI. The peptides can be manufactured by well-controlled chemical protocols and lack toxic and immunogenic impurities [101]. Another issue that still remains challenging for the upscaling process is that rHDL particles which are formed upon self-assembly from peptides, lipids and drugs need subsequent purification and homogenization procedures, which are expansive and laborious.

A major challenge in the synthesis and large-scale production of self-assembled nanostructures like rHDL is the precise control of physicochemical particle properties. The manufacturing technique, composition, concentration, presence of excipients, salts, pH, or temperature might influence the performance of the assembling process. To guarantee batch-to-batch reproducibility between different laboratories, standard operation procedures (SOPs) and defined quality criteria concerning chemical, physical and structural particle parameter need to be elaborated. Criteria for quality control (QC) and quality assurance (QA) are even more strict for cGMP production in pilot plants under clean conditions, which ideally would require in situ quality control measures of the manufacturing process and subsequently for the product stability [171,172,173]. Once established for in vitro applications, another concern regards the in vivo safety, therapeutic dose and efficacy of the rHDL drug delivery system. rHDL particles have to comply with regulatory demands with regard to pharmacokinetic and phamacodynamic profiles. Minimum therapeutically-active doses required to obtain a desired effect, as well as maximal tolerated doses before off-target effects cause toxicity, need to be defined to have a beneficial therapeutic index. Compared to the free drug the delivery system offers the opportunity to accumulate the drug in the target tissue, however, the receptors are also presented in healthy tissues, which could lead to off-target effects and accumulation of the drugs in the liver, lungs, and spleen. Although the rHDL particles are typically non-toxic and well-tolerated, surface modifications or modified chemical compositions might trigger immune response or inflammatory reactions. Accordingly, each new formulation and its single components require individual toxicological assessments, which makes the rHDL drug delivery platform similarly complicated for regulatory issues as nanomedicines based on liposomes or polymers. Safety and pharmacological profiles have to be evaluated separately for each formulation in early stages of clinical development to fulfill the regulatory needs for validated and standardized protocols including in vitro, ex vivo, and in vivo assays [171]. Finally, an optimal clinical trial design has to be chosen to show improved therapeutic efficacy in patients. Despite many challenges the big advantage of rHDL particles for drug delivery is their similarity to native analogs unlike other synthetic materials such as liposomes [24].

## 10. Conclusion and Perspectives

CVDs are among the leading causes of death worldwide and new therapeutic and diagnostic strategies are urgently needed. Lipoproteins offer a very promising nanoparticle platform since (1) the biological roles of endogenous lipoprotein particles in atherosclerosis and CVD are very well known and can be exploited. (2) Lipoproteins, including HDL, offer the unique opportunity to use their intrinsic targeting properties for drug delivery and diagnostics. (3) The structural and compositional features of natural HDL particles can be easily modulated or mimicked. This allows the design of synthetic rHDL particles, which have similar biological functions than their natural counterparts. (4) rHDL particles can be easily manufactured by self-assembling of lipids and proteins/peptides. Different drugs and diagnostic markers can be incorporated into one single multifunctional particle. (5) The small size and well-defined surface properties make rHDL nanocarriers superior to other conventional delivery systems like liposomes for several reasons. In circulation rHDL show less opsonization, longer circulation times, reduced off-target effects, and inherent atheroprotective properties. Enriched with drugs like lipid-lowering compounds or non-coding oligonucleotides synergistic anti-atherosclerotic and anti-inflammatory effects are achieved. The use of fully synthetic rHDL particles engineered by novel microfluidic based manufacturing techniques makes the rHDL platform highly attractive for pharmaceutical companies. However, high manufacturing costs and the fact that rHDL has to be applied as repeated infusion therapy, presently hampers the enthusiasm to use rHDL for the treatment of chronic diseases in which long term medication is needed. Nevertheless, the very positive cardioprotective effects observed for rHDL-based therapies are excellently suited for the management of CVD in patients with an imminent risk for a cardiovascular event at acute phases of atherosclerosis. Based on the current progress in research, it is reasonable to expect that the artificial rHDL platform with its option of tunable architecture, composition and size variability can be optimized in the near future to achieve improved in vivo functions in clinical settings of cardiovascular diseases.

## Figures and Tables

**Figure 1 molecules-24-02829-f001:**
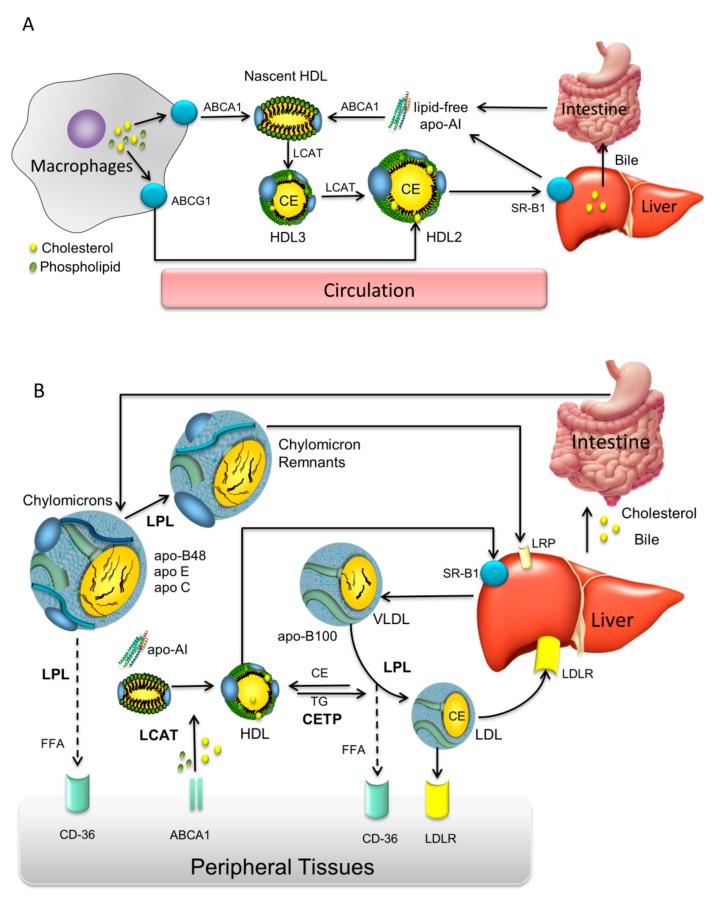
Schematic diagram of high density lipoprotein (HDL) metabolism in the circulatory system. (**A**) apolipoprotein AI (apoA-I) is synthesized in the liver and intestine in lipid free form. apo-AI becomes rapidly lipidated with phospholipids and cholesterol via the hepatocyte ATP-binding cassette A1 (ABCA1) transporter to form discoidal nascent HDL. In peripheral tissues nascent HDL particles take up free cholesterol via the macrophage ABCA1 and ABCG1 transporters. The free cholesterol becomes esterified by lecithin-cholesterol acyltransferase (LCAT) to form cholesteryl esters (CE). Mature HDL3 particles are formed that can be converted to HDL2. Further, HDL2 can be either taken up by the liver via SR-B1 or might be modified by endothelial or hepatic lipases. (**B**) Overview of lipoprotein mediated lipid transport mechanisms. Reprinted with permission from [36].

**Figure 2 molecules-24-02829-f002:**
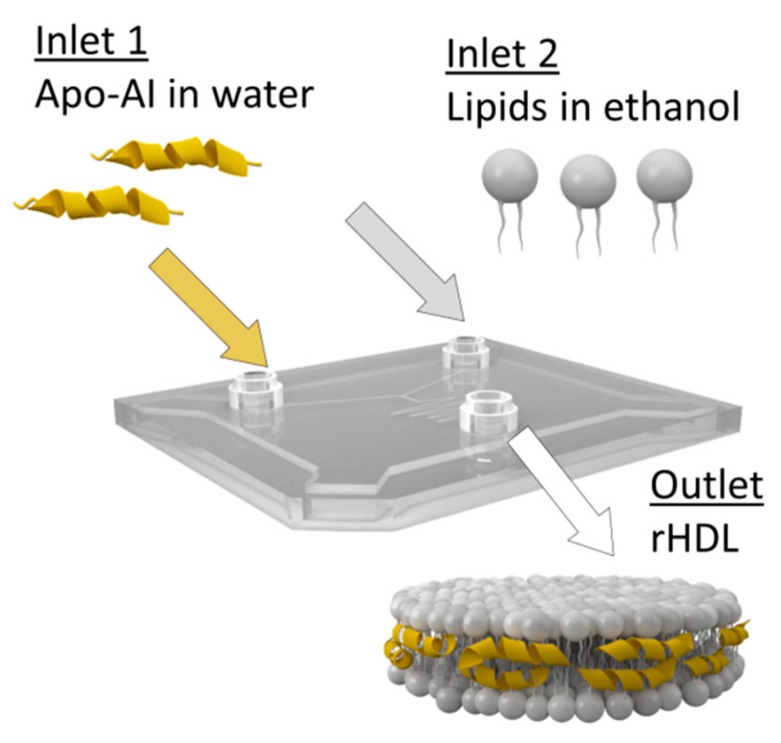
Microfluidic technology for the synthesis of reconstituted high density lipoprotein (rHDL) particles. Using microfluidic devices large amounts of highly homogeneous rHDL nanoparticles can be synthesized in a single step production process.

**Figure 3 molecules-24-02829-f003:**
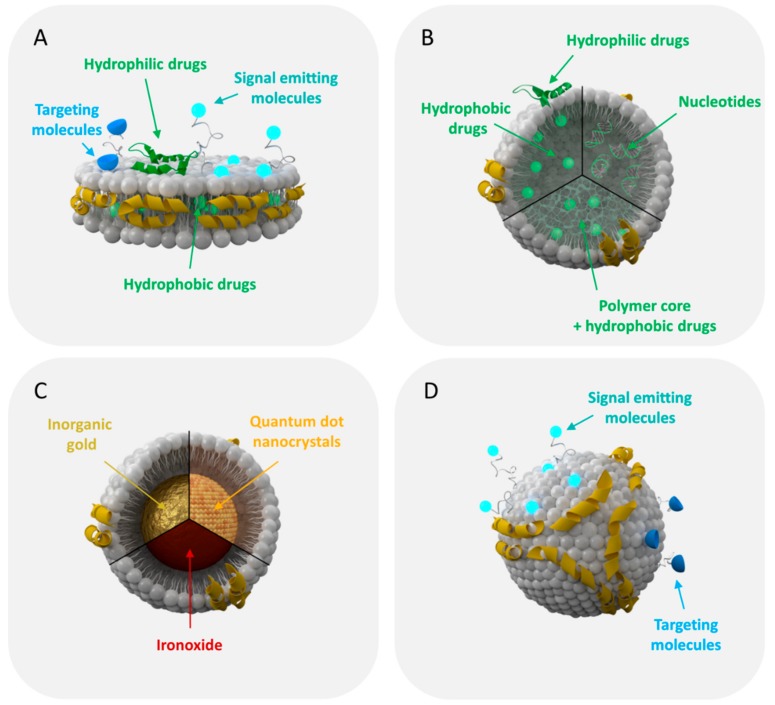
Schematic presentation of the rHDL nanoparticle platform currently explored for drug delivery and imaging. (**A**) apo-AI and phospholipid molecules self-assemble to form rHDL nanodiscs, which can be adapted as multifunctional nanocarrier for drug delivery or imaging. (**B**) Spherical rHDL particles can be loaded with hydrophilic/hydrophobic drugs or oligonucleotides to be used as nanotherapeutics. (**C**) rHDL can be built with an inorganic core of varying size for medical imaging and diagnosis. (**D**) The surface of rHDL can be modified with signal emitting dyes for optical, nuclear, or magnetic resonance imaging. Targeting ligands like antibodies or proteins can coupled to the surface of the particle to redirect rHDL to receptors/biomarkers other than their natural ones.

**Figure 4 molecules-24-02829-f004:**
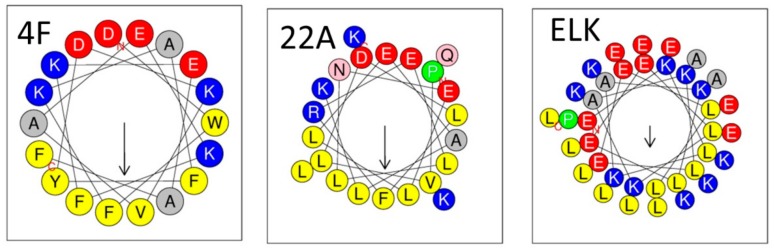
Helical wheel analyses of some of the most prominent representatives of apo-AI mimetic peptides. The peptides belong to class A amphipathic helices [55]. The hydrophobic and hydrophilic residues are grouped on opposing faces with the negatively charged residues being aligned preferentially on the polar phase, while the aromatic residues are rather located on the non-polar face being involved in lipid binding [67]. The helical wheel is obtained with the program HeliQuest [68].

**Figure 5 molecules-24-02829-f005:**
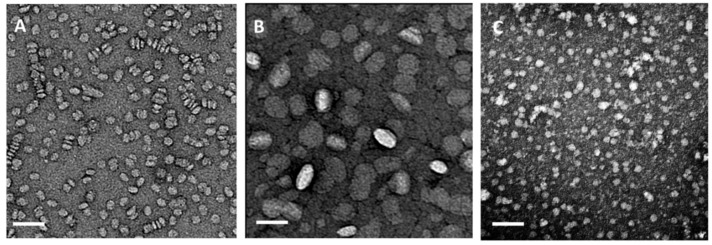
Negative staining transmission electron microscopic (TEM) images of HDL particles. (**A**) rHDL particles reconstituted from dimyristoyl-phosphatidylcholine (DMPC) (8% cholesterol) and full length apo-AI at a molar ratio of 80:1 lipid:protein. The particles are about 10 nm in diameter and discoidal in shape. The circular shapes present nanodiscs viewed from the top. Likewise, stacked nanodiscs resembling “rouleaux” formations are visible. (**B**) rHDL particles assembled from DMPC (8% cholesterol) and apo-AI peptidomimetics (4F) at a molar ratio of 40:1 lipid to peptide. The nanoparticles are larger (about 25–30 nm) and seem to be ellipsoidal in shape; (**C**) HDL particles isolated from human plasma are spherical with particles sizes in the range from 8 nm to 14 nm (unpublished data). The bar represents 50 nm.

**Figure 6 molecules-24-02829-f006:**
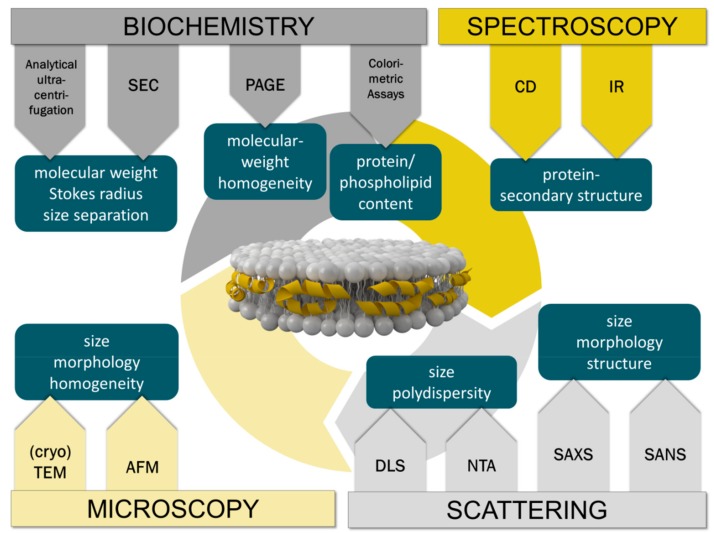
Overview of the physicochemical techniques commonly used for the characterization of rHDL particles. The methods are grouped into four main categories indicating the information obtained from each technique.

**Figure 7 molecules-24-02829-f007:**
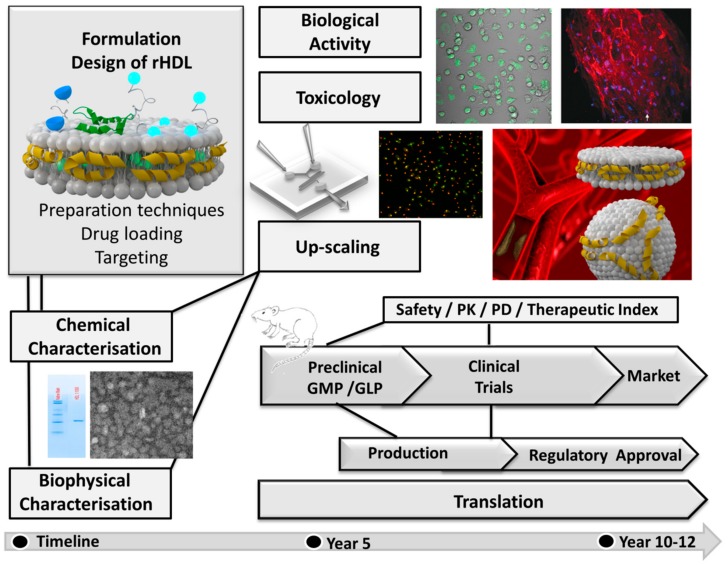
Simplified scheme showing a tentative time frame for the development of rHDL based drug delivery systems depicting the steps involved in translational research from bench to bedside.

**Table 1 molecules-24-02829-t001:** Representative examples of various rHDL nanoparticles designed for the therapy of CVD.

Composition	Drug	Preclinical/Clinical	Activity/Primary Outcome	Ref.
**rHDL Based Therapy in CVDs**				
DPPC/purified human apo-AI		normocholesterolemic rabbit acute arterial inflammation model	neutrophil infiltration↓ ROS↓ adhesion molecules ↓	[86]
CSL-111: SoyPC/purified human apo AI (150/1 molar ratio)		patients with acute coronary syndromes	atheroma volume↓	[90]
CSL-112: SoyPC/purified human apo AI (75/1 molar ratio)		patients with stable atherothrombotic disease Phase 2a	apo-AI ↑ cholesterol efflux capacity↑	[91]
CER-001: recombinant apoAI/Egg SM/DPPG (1/2.7/0.1 molar ratios)		patients with carotid artery disease	apo-AI ↑ cholesterol efflux capacity↑	[96]
**rHDL Drug Delivery**				
Lyso-PC/DMPC/purified human apo-AI	simvastatin	apo E-/- mouse model	inflammation ↓	[135]
SoyPC/cholesterol/purified apo-AI (disc)	tanshinone IIA	atherosclerotic NZW rabbit	cholesterol efflux capacity↑	[136]
SoyPC/cholesterol/cholesteryl oleate/glycerol trioleate (spheres)	tanshinone IIA	atherosclerotic NZW rabbit	cholesterol efflux capacity↑	[136]
**rHDL apo-AI Mimetic Peptide**				
POPC/5A (7/1 molar ratio)		Sprague-Dawley rats apo E-/- mouse model	cholesterol efflux capacity↑ atheroma area ↓	[156]
SM/5A (7/1 molar ratio)		Sprague-Dawley rats apo E-/- mouse model	cholesterol efflux capacity↑ atheroma area ↓	[156]
DMPC/branched, multivalent peptides		LDLr −/− mouse model	plasma total cholesterol levels ↓	[157]
ETC-642: 22A/SM/DPPC (1/1/1 molar ratios)		atherosclerotic NZW rabbit	ICAM ↓, VCAM ↓ vascular inflammation↓ oxLDL↓	[159]

**Abbreviations:** DPPC: dipalmitoyl- phosphatidylcholine; PS: phosphatidylserine; SoyPC: soybean phosphatidylcholine; SM: sphingomyelin; DPPG: dipalmitoyl-sn-glycero-3-phosphorylglycerol; LysoPC: 1-Myristoyl-2-Hydroxy-sn-Glycero-3-Phosphatidylcholine; DMPC: dimyristoyl-phosphatidylcholine; LDLr: low density lipoprotein receptor; ICAM: intracellular cell adhesion molecule-1; VCAM: vascular cell adhesion molecule-1; oxLDL: oxidized low density lipoprotein.

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
