# Peer review of "Artificial High Density Lipoprotein Nanoparticles in Cardiovascular Research"

_molecules, 2019, doi:10.3390/molecules24152829_

Round 1
Reviewer 1 Report
The review paper entitled Artificial High Density Lipoprotein Nanoparticles in Cardiovascular Research systematically summarize the state-of-art development of this field and provide a broad view of biomedical uses of lipoprotein, which will benefit the later researchers. Overall, the manuscript is in excellent shape and well organized. I am more than happy to recommend its publication on the journal of Molecules.
Author Response
Reviewer 1:
The review paper entitled Artificial High Density Lipoprotein Nanoparticles in Cardiovascular Research systematically summarize the state-of-art development of this field and provide a broad view of biomedical uses of lipoprotein, which will benefit the later researchers. Overall, the manuscript is in excellent shape and well organized. I am more than happy to recommend its publication on the journal of Molecules.
We would like to thank the reviewer for his/her overall very positive comments on our paper.
Reviewer 2 Report
This paper is a very good and intensive review on HDL and its use in cardiovascular research. The review summarizes the state-of-the-art of lipoprotein metabolism for cardiovascular disease, published technologies for the production of rHDLs, and structural aspects of apo-AI mimetic peptides discussed in the literature. The text and also the selected representations are, in my estimation, very complete and very good to follow. Only in Figure 3 I would like to clarify the color code used.
In the following sections, however, the authors try to do without further illustrations or representations. In my opinion, this leads to a significantly reduced readability of the review. The section on physical-chemical characterization of rHDLs is certainly not incomplete with regard to the enumeration of relevant methods, but one might wish for a more in-depth discussion of the chemical characterization of rHDLs. The discussion on the advantages and disadvantages of the methods will not be completed. For example, the disadvantages of NTA regarding up-scaling and statistics remain unexplained. SEC and PAGE certainly have not only the disadvantage of not distinguishing between spherical and discoidal particles. Here I would have liked corresponding real pictures of the rHDL's. Corresponding pictures should be added.
The very comprehensive section on the therapeutic possibilities of the rHDL's testifies to the extremely extensive research work of the authors and represents the currently examined and published approaches lyrically very well. However, in this section completely on illustrations, schematics, tables or images omitted. Not only does this reduce the enjoyment of reading the review, it also makes the entire review less readable. I recommend here the supplement of pictures.
The summary and discussion of the challenges, limitations and benefits of rHDL has been outstanding. I would have preferred only a more in-depth discussion on the scale-up of manufacturing processes and the necessary quality control that follows from the previous discussion on chemical-physical characterization.
Overall, the review is a very good and important work of the authors and I recommend the publication of the paper in molecules. However, I recommend before the publication, the second half of the review to supplement appropriate illustrations.
